# Bottom-up and top-down neural signatures of disordered multi-talker speech perception in adults with normal hearing

Aravindakshan Parthasarathy[1,2]*, Kenneth E Hancock[1,2], Kara Bennett[3], Victor DeGruttola[4], Daniel B Polley[1,2]

[1]Eaton-Peabody Laboratories, Massachusetts Eye and Ear Infirmary, Boston, United States; [2]Department of Otolaryngology – Head and Neck Surgery, Harvard Medical School, Boston, United States; [3]Bennett Statistical Consulting Inc, Ballston, United States; [4]Department of Biostatistics, Harvard TH Chan School of Public Health, Boston, United States

**Abstract** In social settings, speech waveforms from nearby speakers mix together in our ear canals. Normally, the brain unmixes the attended speech stream from the chorus of background speakers using a combination of fast temporal processing and cognitive active listening mechanisms. Of >100,000 patient records,~10% of adults visited our clinic because of reduced hearing, only to learn that their hearing was clinically normal and should not cause communication difficulties. We found that multi-talker speech intelligibility thresholds varied widely in normal hearing adults, but could be predicted from neural phase-locking to frequency modulation (FM) cues measured with ear canal EEG recordings. Combining neural temporal fine structure processing, pupil-indexed listening effort, and behavioral FM thresholds accounted for 78% of the variability in multi-talker speech intelligibility. The disordered bottom-up and top-down markers of poor multi-talker speech perception identified here could inform the design of next-generation clinical tests for hidden hearing disorders.

**\*For correspondence:**
Aravindakshan_Parthasarathy@
meei.harvard.edu

**Competing interest:** See
page 16

**Reviewing editor:** Huan Luo,
Peking University, China

## Introduction

Slow fluctuations in the sound pressure envelope are sufficient for accurate speech perception in quiet backgrounds (*Shannon et al., 1995*). Envelope cues are less useful when speech is embedded in fluctuant backgrounds comprised of multiple talkers, environmental noise or reverberation (*Zeng et al., 2005*). Under these conditions, segregating a target speech stream from background noise requires accurate encoding of low-frequency spectral and binaural cues contained in the stimulus temporal fine structure (sTFS) (*Hopkins and Moore, 2009*; *Lorenzi et al., 2006*). Monaural sTFS cues convey acoustic signatures of target speaker identity based on the arrangement of peaks in the sound spectrum (e.g., formant frequencies of target speech), while binaural sTFS cues can support spatial separation of target and competing speakers via interaural phase differences (*Moore, 2014*). With aging and hearing loss, monaural and binaural sTFS cues become less perceptually available, even when audibility thresholds for low-frequency signals that convey sTFS cues are normal (*Buss et al., 2004*; *DiNino et al., 2019*; *Füllgrabe et al., 2014*; *Léger et al., 2012*; *Lorenzi et al., 2009*; *Mehraei et al., 2014*; *Moore, 2014*; *Strelcyk and Dau, 2009*). The biological underpinnings for poor sTFS processing with aging or hearing impairment are unknown, but may reflect the loss of auditory nerve afferent fibers, which degenerate at the rate of approximately 1000 per decade, such that only half survive by the time a typical adult has reached 40 years of age (*Makary et al., 2011*;

**eLife digest** Our ears were not designed for the society our brains created. The World Health Organization estimates that a billion young adults are at risk for hearing problems due to prolonged exposure to high levels of noise. For many people, the first symptoms of hearing loss consist in an inability to follow a single speaker in crowded places such as restaurants.

However, when Parthasarathy et al. examined over 100,000 records from the Massachusetts Eye and Ear audiology database, they found that around 10% of patients who complained about hearing difficulties were sent home with a clean bill of hearing health. This is because existing tests do not detect common problems related to understanding speech in complex, real-world environments: new tests are needed to spot these hidden hearing disorders. Parthasarathy et al. therefore focused on identifying biological measures that would reflect these issues.

Normally, the brain can 'unmix' different speakers and focus on one person, but even in the context of normal hearing, some people are better at this than others. Parthasarathy et al pinpointed several behavioral and biological markers which, when combined, could predict most of this variability. This involved, for example, measuring the diameter of the pupil while people are listening to speech in the presence of several distracting voices (which mirrors how intensively they have to focus on the task) or measuring the participants' ability to detect subtle changes in frequency (which reflects how fast-changing sound elements are encoded early on in the hearing system). The findings show that an over-reliance on high-level cognitive processes, such as increased listening effort, coupled with problems in the early processing of certain sound traits, was associated with problems in following a speaker in a busy environment.

The biological and behavioral markers highlighted by Parthasarathy et al do not require specialized equipment or marathon sessions to be recorded. In theory, these tests could be implemented into most hospital hearing clinics to give patients and health providers objective data to understand, treat and monitor these hearing difficulties.

*Wu et al., 2019*). A selective loss of cochlear afferent fibers would not likely affect audibility thresholds, but could adversely affect the ability of the auditory system to fully exploit suprathreshold monaural and binaural sTFS cues that are critical for multi-talker speech intelligibility (*Deroche et al., 2014*; *Hopkins et al., 2008*; *Jin and Nelson, 2010*; *Lopez-Poveda and Barrios, 2013*; *Moore and Glasberg, 1987*; *Qin and Oxenham, 2003*, for review see - *Moore, 2014*).

Accurate processing of a target speaker in a multi-talker background reflects a harmony between high-fidelity encoding of bottom-up acoustic features such as sTFS alongside cognitive signatures of active listening including attention, listening effort, memory, multisensory integration and prediction (*Best et al., 2009*; *Gordon-Salant and Cole, 2016*; *Narayan et al., 2007*; *Pichora-Fuller et al., 2016*; *Wild et al., 2012*; *Winn et al., 2015*). These top-down assets can be leveraged to compensate for poorly resolved bottom-up sensory cues, suggesting that listeners with clinically normal hearing that struggle to process speech in noise might be identified by an over-reliance on top-down active listening mechanisms to de-noise a corrupted afferent speech input (*Besser et al., 2015*; *Ohlenforst et al., 2017*; *Winn et al., 2015*). Here, we apply parallel psychophysical and neurophysiological tests of sTFS processing in combination with physiological measures of effortful listening to converge on a set of neural biomarkers that identify poor multi-talker speech intelligibility in adults with clinically normal hearing.

## Results

### Many individuals seek medical care for poor hearing but have no evidence of hearing loss

We identified the first visit records of English-speaking adult patients from the Massachusetts Eye and Ear audiology database over a 16 year period, with complete bilateral audiometric records at six octave frequencies from 250 Hz to 8000 Hz according to the inclusion criteria in *Figure 1A*. Of the 106,787 patient records that met these criteria, we found that approximately one out of every five patients had no clinical evidence of hearing loss, defined as thresholds > 20 dB HL at test

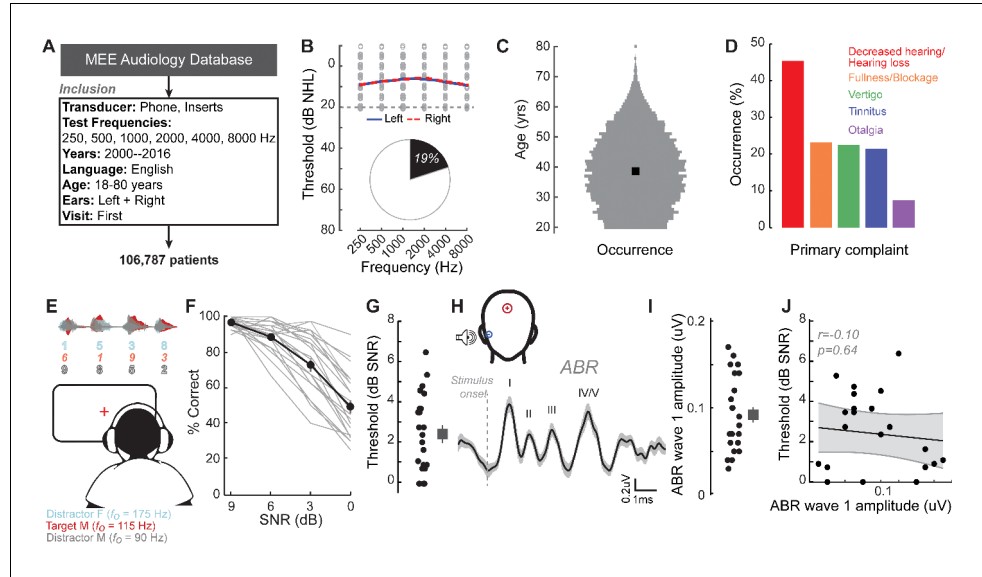

**Figure 1.** A normal audiogram does not guarantee robust speech intelligibility in everyday listening conditions. (**A**) Screening criteria for eligible audiology patient records from our hospital collected between 2000 and 2016. (**B**) Bilateral normal audiograms, defined as thresholds better than 20 dB HL (gray dashed line) were identified in 19% of the total patient population. Average audiograms from the left (blue) and right (red) ears are shown with individual data points in gray open circles. (**C**) Normalized age distribution of patients with bilateral normal audiograms shows a larger percentage of younger and middle-aged patients between 20–50 years of age. Black square indicates median age of 39 years. (**D**) Top five primary complaints that resulted in the visit to the clinic for these patients, including perceived hearing loss or decreased hearing presenting in 45% of these patients. (**E**) Schematic of a multi-talker digit recognition task. Subjects (N = 23) were familiarized with a target male speaker (red) producing four digits between 1 and 9 (excluding the bi-syllabic '7'), while two spatially co-localized distractors, one male and one female, with $F_0$ frequencies above and below the target speaker simultaneously spoke four digits at varying signal-to-noise ratios (SNRs). (**F**) Accuracy decreased as a function of SNR at variable rates and to variable degrees. Correct trials required correctly reporting all four digits. (**G**) Variability in individual speech reception thresholds, defined as the SNR that produced a 70.7% success rate. Value at right represents sample mean ± SEM. (**H**) Auditory brainstem responses measured using ear canal tiptrodes yielded robust wave one amplitudes, a marker for auditory nerve integrity. Data reflect mean ± SEM. (**I**) Wave one values from individual subjects (left) and mean ± SEM of the sample (right). (**J**) No significant associations were observed between the ABR wave one amplitudes and speech reception threshold on the multi-talker digit task. r = Pearson's correlation, and shaded area indicates 95% confidence intervals of the regression line (black) in *Figures 1–4*.

The online version of this article includes the following source data and figure supplement(s) for figure 1:

**Source data 1.** Digits comprehension thresholds and ABR wave one amplitudes.
**Figure supplement 1.** Audiometric characteristics of patients with normal audiograms that present at the Massachusetts Eye and Ear audiology clinic with complaints of poor hearing.
**Figure supplement 2.** Audiometric profiles and markers of noise exposure in study participants.
**Figure supplement 2—source data 1.** High-frequency audiometry and noise exposure questionnaire values.
**Figure supplement 3.** Experimental study design.
**Figure supplement 4.** Digits comprehension task captures aspects of self-reported difficulties in real-world multi-talker listening conditions experienced by the participants.
**Figure supplement 4—source data 1.** Mean values from the SSQ questionnaire.

frequencies up to 8 KHz (19,952, 19%, *Figure 1B*). The majority of these individuals were between 20–50 years old (*Figure 1C*) and had no conductive hearing impairment, nor focal threshold shifts or 'notches' in their audiograms greater than 10 dB (*Figure 1—figure supplement 1A*). The thresholds between their left and right ears were also symmetrical within 10 dB for >95% of these patients (*Figure 1—figure supplement 1B*). Despite their clean bill of hearing health, 45% of these individuals primarily complained of decreased hearing or hearing loss (*Figure 1D*). Absent any objective

measure of hearing difficulty, these patients are typically informed that their hearing is 'normal' and that they are not expected to experience communication problems.

## Speech-in-noise intelligibility varies widely in individuals with clinically normal hearing

Our database analysis suggested that approximately one in ten adults arrived to our clinic seeking care for reduced hearing, only to be told that their hearing was fine. This was not entirely surprising, as most clinical tests are not designed to capture difficulties with 'real world' speech communication problems that likely prompted their visit to the clinic. To better understand the nature of their supra-threshold hearing problems, we recruited 23 young or middle-aged listeners (mean age: 28.3 ± 0.9 years) that matched the clinically normal hearing from the database profile (*Figure 1—figure supplement 2A*). Our subjects agreed to participate in a multi-stage research study consisting of self-reported questionnaires, behavioral measures of hearing, and EEG measures of auditory processing (*Figure 1—figure supplement 3*).

Like the patients from the clinical database, the audiograms from these subjects were clinically normal, yet many reported difficulties with speech intelligibility, particularly in listening conditions with multiple overlapping speakers (*Figure 1—figure supplement 4A*). We directly measured speech-in-noise intelligibility with a digits comprehension task, which simulates the acoustic challenge of focused listening in multi-talker environments, while eliminating linguistic and contextual speech cues. Subjects attended to a familiar male speaker ($F_0$ = 115 Hz) producing a stream of four digits in the presence of one male and one female distracting speakers ($F_0$ = 90 Hz and 175 Hz, respectively). The distracting speakers produced digits simultaneously at variable signal-to-noise ratios (SNRs) (*Figure 1E*). Performance on the digits task decreased as a function of SNR (*Figure 1F*). Speech reception thresholds, defined as the 70.7% correct point on the response curve, varied widely across a 0–7 dB SNR range with a mean of 2.42 dB (*Figure 1G*). We found that speech intelligibility thresholds were significantly correlated with the subjects' self-reported difficulties in multi-speaker conditions, suggesting that the digits comprehension task captures aspects of their real-world communication difficulties (r = 0.46, p=0.02, *Figure 1—figure supplement 4B*).

## Peripheral markers of noise damage do not explain performance on the speech-in-noise task

We first determined whether simple adaptations of existing clinical tests could identify deficits in multi-talker speech intelligibility. We measured hearing thresholds at extended high frequencies, a marker for early noise damage (*Fausti et al., 1981*; *Le Prell et al., 2013*; *Mehrparvar et al., 2011*). Subjects exhibited substantial variability in their extended high frequency thresholds (>8 kHz) despite having clinically normal audibility at lower frequencies (*Figure 1—figure supplement 2B*). We also measured the amplitude of auditory brainstem response (ABR) wave 1, which can reveal age- or trauma-related changes in auditory nerve health (*Figure 1H–I*) (*Fernandez et al., 2015*; *Liberman et al., 2016*; *Parthasarathy and Kujawa, 2018*). ABR wave one amplitude and extended high frequency thresholds both showed substantial variability in subjects with clinically normal audiograms, but neither could account for performance on the competing digits task (r = 0.10 p=0.64, *Figure 1J*, *Figure 1—figure supplement 2D*).

## Encoding of sTFS cues predicts speech-in-noise intelligibility

Poor processing of sTFS cues has long been associated with elevated speech recognition thresholds, especially in patients with hearing loss (*Lorenzi et al., 2006*). In a classic test of sTFS processing, subjects were asked to detect a slow, subtle FM imposed on a low-frequency carrier (*King et al., 2019*; *Moore and Sek, 1996*; *Moore and Sek, 1995*; *Moore and Skrodzka, 2002*; *Sek and Moore, 1995*; *Wallaert et al., 2018*). We tested our subjects with this psychophysical task, which uses an adaptive two-interval forced choice procedure to converge on the threshold for detecting FM of a 500 Hz tone (*Figure 2A*). FM detection thresholds varied widely between subjects (*Figure 2B*) and were strongly correlated with performance on the competing digits task (r = 0.85, p<0.001, *Figure 2C*). We were struck that detection thresholds for such a simple stimulus could accurately predict performance in a much more complex task. On the one hand, sensitivity to FM could reflect superior low-level encoding of sTFS cues that are critical for segregating a target speech stream

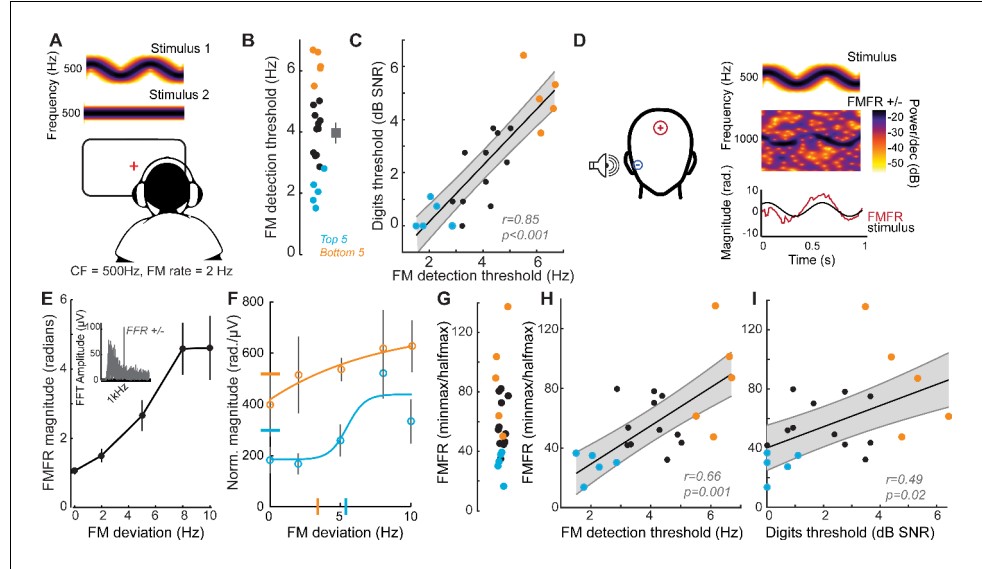

**Figure 2.** Perceptual and neural processing of sTFS cues predict speech in noise intelligibility. (**A**) Design of a psychophysical task to measure frequency modulation (FM) detection threshold. Participants indicated FM in a 500 Hz tone in an adaptive (2-down 1-up) two alternative forced choice task. (**B**) Individual (left) and average ± SEM (right) FM detection thresholds. Top and bottom five performers (~20th percentile) on the behavioral FM detection task are shown in blue and orange respectively, in panels B-C and F-I (**C**) FM detection thresholds were strongly predictive of speech in noise recognition threshold defined with the multi-talker digit comprehension task. (**D**) An objective neurophysiological measure of monaural sTFS processing was obtained using ear canal (-) and scalp (+) electrodes, and a 500 Hz tone presented with various FM deviations in alternating polarity. The averaged response was analyzed at 1000 Hz (2F) in order to minimize contributions by the cochlear microphonic and emphasize neural generators. The magnitude of the FM following response (FMFR) was computed using a heterodyne. (**E**) The FMFR magnitude increased as a function of FM deviation up to ~8 Hz. *Inset:* The FMFR magnitude was normalized by the pure tone phase-locking amplitude of each subject to minimize variability due to head size and recording conditions. (**F–G**) A sigmoidal fit to the normalized FMFR growth curve was used to calculate an FMFR measure of slope for each subject, by dividing the overall dynamic range of the response by the halfway point to the maximum (illustrated in (**F**) for the top and bottom five performers of the behavioral task). Blue and orange bars indicate the X and Y axes intercepts of the halfway point of the fit. (**H–I**) The neurophysiological FMFR was strongly predictive of FM detection thresholds determined with the psychophysical task (**H**) as well as performance on the digits comprehension task (**I**).

The online version of this article includes the following source data and figure supplement(s) for figure 2:

**Source data 1.** FM detection thresholds and FMFR slope values.
**Figure supplement 1.** Determination of optimal electrode montages for obtaining electrophysiological responses.
**Figure supplement 1—source data 1.** ABR wave 1 – wave 5 indices for various electrode montages.
**Figure supplement 2.** Cortical event-related potentials (ERPs) are modulated by FM stimuli, but not related to behavioral performance.
**Figure supplement 2—source data 1.** ERP measures.

from distractors. Alternatively, perceptual thresholds for FM could reflect a superior abstracted representation of stimulus features at any downstream stage of neural processing, and not the high fidelity representation of sTFS cues, per se. Taking this line of argument a step further, a correlation between competing talker thresholds and FM thresholds may not reflect the stimulus representation at all, but instead could reflect subjects' general aptitude for utilizing cognitive resources such as attention and effort to perform a wide range of listening tasks (*Pichora-Fuller et al., 2016*).

To test the hypothesis that early neural processing of sTFS cues in the FM tone is associated with superior speech-in-noise processing, we developed a non-invasive physiological measure of neural sTFS phase-locking at early stages of auditory processing. We first determined that an ear canal to Fz electrode montage was sensitive to evoked potentials generated by the auditory nerve (*Figure 1I*, *Figure 2—figure supplement 1A–B*), but we also wanted to exclude any pre-neural

contributions, such as the cochlear microphonic, that are generated by hair cells (*Fettiplace, 2017*). Because pre-neural responses are nearly sinusoidal for low-frequency tones, but auditory nerve fiber discharges are partially half-wave-rectified, we could isolate the neural component by alternating the polarity of FM tones, averaging the responses, and analyzing the phase-locking at twice the carrier frequency (*Lichtenhan et al., 2014*). We observed robust phase-locked following response to the FM stimulus (*Figure 2D*, termed the FM following response or FMFR). We used a heterodyne method to extract the FMFR for FM depths up to 10 Hz, or ~0.02 octaves (*Figure 2E*). To factor out variability in phase-locking due to head size and overall electrode SNR we calculated the amplitude of the carrier frequency following response to a tone with 0 Hz FM and then expressed the FMFR magnitude as a fraction of this value (*Figure 2E*, inset). Sigmoidal fits to the FMFR growth function (illustrated for the top and bottom five performers on the behavioral task in *Figure 2F*) were further reduced to a single value per subject by dividing the maximum dynamic range for each subject (min-max) by the halfway point to get a measure of slope (halfmax; *Figure 2G*). With this approach, subjects with a wide dynamic range for encoding FM depth have more robust FM encoding and therefore smaller min-max/halfmax ratio values.

We found that robust low-level encoding of FM cues was highly predictive of an individual's performance on the FM psychophysical detection task (r = 0.66 p=0.001; *Figure 2H*), suggesting that the FMFR can provide an objective neurophysiological measure of an individual's ability to encode sTFS cues. Importantly, the FMFR was also significantly correlated with thresholds on the competing digits task (r = 0.49, p=0.02, *Figure 2I*). Whereas phase-locking to the sTFS cues in the FM tone was related to psychophysical performance and speech recognition thresholds, the cortical evoked potentials recorded simultaneously from the same stimuli were not correlated with either measure (*Figure 2—figure supplement 2*). These data suggest that the strong association between psychophysical tests for FM detection and speech-in-noise intelligibility can be attributed, at least in part, to encoding of FM cues at early stages of auditory processing.

## Encoding of unrelated sTFS cues do not predict speech-in-noise intelligibility

We reasoned that the correlation between low-level FM encoding and speech intelligibility might just reflect a correlation between any measure of fast temporal processing fidelity and speech intelligibility. This could be addressed by measuring temporal processing fidelity on an unrelated stimulus and noting whether it had any correlation with speech-in-noise thresholds. Interaural timing cues can improve speech processing in noise, but would not be expected to have any association with the competing digits task used here, where the identical waveform was presented to both ears. To test whether poor encoding of binaural sTFS cues would also predict poor performance in the competing digits task, we performed parallel psychophysical and electrophysiological measurements of sensitivity to interaural phase differences (*Haywood et al., 2015*; *McAlpine et al., 2016*; *Ross et al., 2007*; *Undurraga et al., 2016*).

In this task, the phase of a 520 Hz tone presented to each ear was shifted by a variable amount (up to 180°), creating the percept of a tone that moved from the center to the sides of the head. To eliminate phase transition artifacts, an amplitude modulation of ~41 Hz was imposed on the tone, such that the instantaneous phase shift always coincided with the null of the amplitude envelope (*Figure 3A*) (*Haywood et al., 2015*; *Undurraga et al., 2016*). To test psychophysical thresholds for binaural sTFS cues, subjects indicated the presence of an interaural phase difference (IPD) in one of two tokens presented in a 2-interval forced choice task. IPD thresholds were variable, ranging between 5 and 25 degrees (*Figure 3B*). To quantify electrophysiological encoding of IPD, recordings were made with electrodes in a vertical Fz-C7 montage to emphasize binaural generators (*Figure 2—figure supplement 1B*). The IPD was alternated at 6.8 Hz, inducing a following response to the IPD (IPDFR) as well as a phase-insensitive envelope following response at the 41 Hz amplitude modulation rate (*Figure 3C*). As expected, the amplitude of the IPDFR at 6.8 Hz increases monotonically with larger interaural time differences, whereas the amplitude of the envelope following response remains constant (*Figure 3D*). As above, we minimized variability due to head size and electrode SNR by expressing the IPDFR amplitude as a fraction of the envelope following response. Sigmoidal fits to the normalized growth curves were then used to calculate min-max/halfmax values, similar to the FMFRs (shown for the top and bottom five performers on the behavioral task in *Figure 3E*). Like the FMFR above, we noted a strong association between an individual's psychophysical threshold

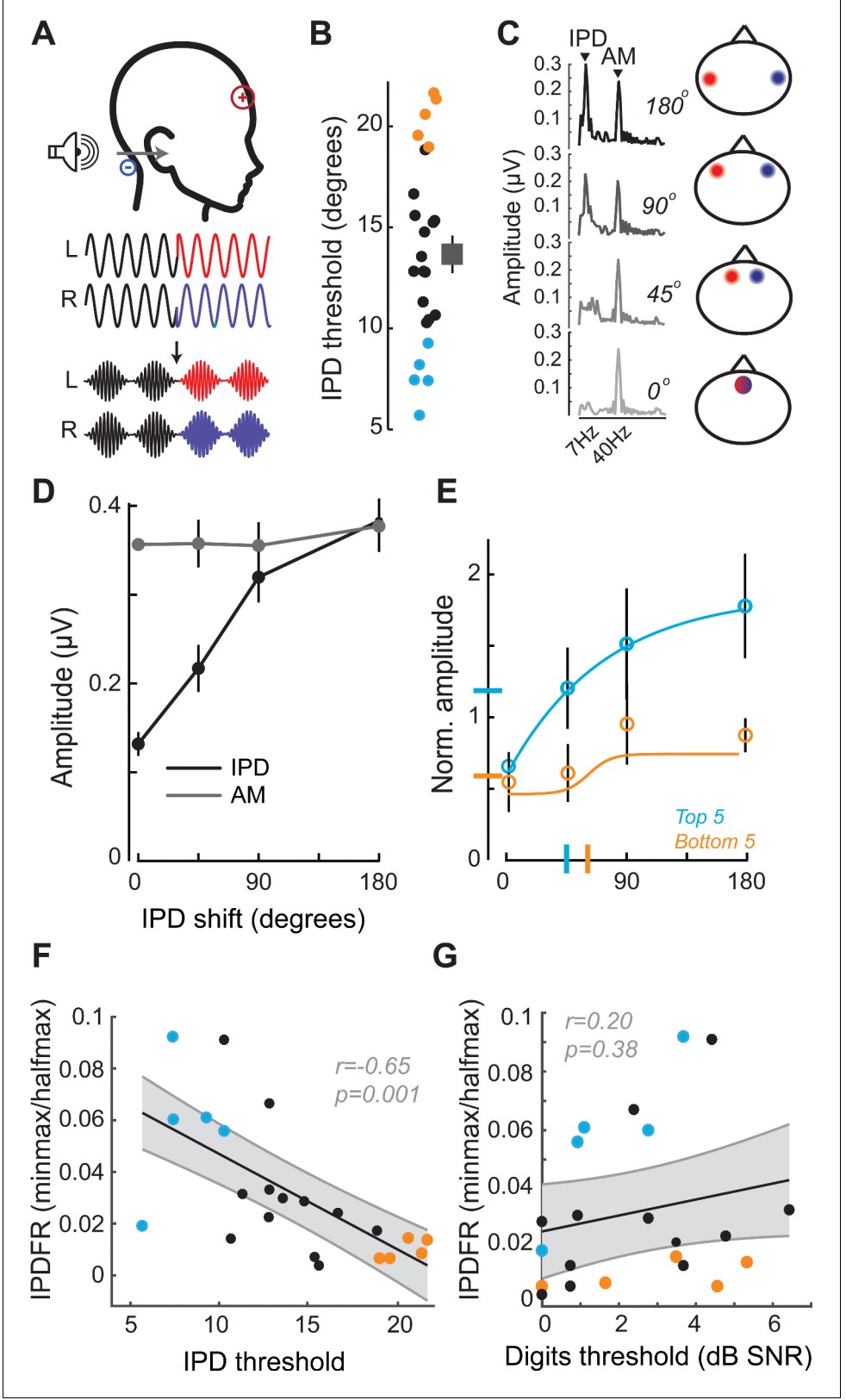

**Figure 3.** Neural and perceptual processing of rapid temporal cues unrelated to the speech task do not predict speech recognition thresholds. (**A**) Design of interaural phase difference (IPD) detection task. The phase of a 520 Hz tone instantaneously shifted from diotic (aligned interaural phase) to dichotic (variable interaural phase).
*Figure 3 continued on next page*

*Figure 3 continued*

Amplitude modulation (AM) at 40.8 Hz was aligned to the interaural phase shift such that the amplitude minimum coincided with the phase transition. (**B**) Minimal IPD detection threshold was measured in a 2-alternative forced choice task. IPD thresholds varied between 5 and 25 degrees across individual subjects (left), mean ± SEM shown at right. Top and bottom five performers on the behavioral FM detection task are shown in blue and orange respectively. (**C–D**) In EEG recordings, the IPD alternated between diotic and dichotic at a rate of 6.8 Hz. Fast Fourier transforms of scalp-recorded evoked responses revealed a phase-dependent IPD following response (IPDFR) at 6.8 Hz and a 40.8 Hz AM envelope following response. (**E**) The IPDFR magnitude was expressed as a fraction of the envelope following response for each subject to minimize variability due to head size and recording conditions. Min-max and half-max values were computed from sigmoidal fits to the normalized IPDFR growth function (Illustrated here for the top and bottom five performers on the behavioral task) (**F–G**) The IPDFR was strongly predictive of IPD detection thresholds (**F**), but not performance on the digits comprehension task (**G**). The online version of this article includes the following source data for figure 3:

**Source data 1.** IPD detection thresholds and IPDFR slope values.

for IPD and the growth of the electrophysiological IPDFR (r = −0.65 p=0.001, *Figure 3F*). Unlike the FMFR, subjects that were most sensitive to IPD showed a large, rapid increase in IPDFR amplitude across the testing range, resulting in a large min-max and a small half-max (*Figure 3E*). As a result, the correlation between psychophysical threshold and IPDFR is negative (*Figure 3F*) whereas the correlation between FM threshold and the FMFR amplitude is positive (*Figure 2H*). This can be attributed to inherent differences in the FMFR (a measure of sTFS phase-locking) versus the IPDFR (a measure of sTFS envelope processing; see Discussion). More to the point, neither the IPD psychophysical threshold, nor the IPDFR amplitude had statistically significant correlations with the digits in noise threshold, confirming that task performance was specifically linked to encoding of task-relevant FM cues and not a general sensitivity to unrelated sTFS cues (IPD threshold and speech, r = 0.21 p=0.34; IPDFR and speech, r = 0.20 p=0.38, *Figure 2G*).

## Pupil-indexed effortful listening predicts speech intelligibility

Speech recognition is a whole brain phenomenon that is intimately linked to cortical processing as well as cognitive resource allocation such as listening effort, spatial attention, working memory, and prediction (*Ding and Simon, 2012*; *Mesgarani and Chang, 2012*; *O'Sullivan et al., 2015*; *Peelle, 2018*; *Ruggles et al., 2011*; *Shinn-Cunningham et al., 2017*; *Song et al., 2014*). In this sense, encoding of bottom-up sTFS cues can provide critical building blocks for downstream speech processing but ultimately provide an incomplete basis for predicting performance on cognitively demanding listening tasks. To capture variability in speech processing that was not accounted for by sTFS cues, we measured task-evoked changes in pupil diameter, while subjects performed the digits comprehension task. Under isoluminous conditions, pupil diameter can provide an objective index of the sensory and cognitive challenge of processing a target speech stream in the presence of distracting speakers (*Koelewijn et al., 2015*; *Wang et al., 2018*; *Winn et al., 2015*; *Zekveld et al., 2010*). Prior work has shown that increased pupil dilation in low SNR listening conditions can reflect greater utilization of top-down cognitive resources to enhance attended targets, whereas smaller pupil changes have been associated with higher fidelity bottom-up inputs that do not demand additional listening effort to process accurately (*Koelewijn et al., 2012*; *Zekveld et al., 2014*; *Zekveld and Kramer, 2014*) (*Figure 4A*).

We confirmed here that the fractional change in pupil diameter was linearly related to the SNR of the target speaker (*Figure 4B*) in 16 subjects that provided measurable pupil signals (see Materials and methods for a statement on exclusion criteria). In the same spirit as removing variability related to head size and electrode SNR, we first factored out unrelated measurement noise by expressing the SNR-dependent change in pupil diameter as a fraction of light-induced pupil change in each subject (*Figure 4C*). We confirmed that individuals with steeper pupil recruitment functions had more difficulty in the multi-talker speech task, leading to a significant correlation between min-max/halfmax pupil change and speech intelligibility threshold (r = 0.53, p=0.03, *Figure 4D*).

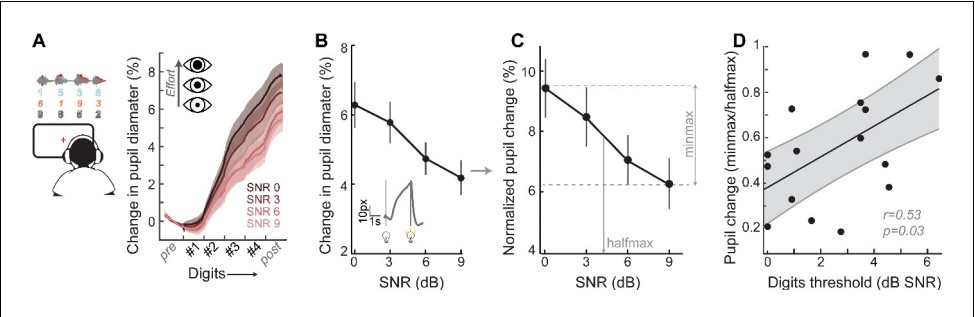

**Figure 4.** A pupil-indexed measure of effortful listening predicts multi-talker speech recognition thresholds. (**A**) Fractional change in pupil diameter was measured under isoluminous conditions before, during and after the 4-digit sequence at various SNRs. (**B**) The peak fractional change in pupil diameter was normalized to the light-induced pupil change for each SNR (**C**). The SNR-dependent change in pupil diameter was calculated as the min-max/halfmax. (**D**) Greater recruitment of pupil-indexed effortful listening across SNRs was significantly associated with the speech intelligibility threshold. Baseline changes in pupil across the testing session, taken as a measure of listening fatigue, showed no relationship with task performance (*Figure 4—figure supplement 1*).

The online version of this article includes the following source data and figure supplement(s) for figure 4:

**Source data 1.** Pupil diameter slope values.
**Figure supplement 1.** Listening fatigue does not account for performance on the multi-talker digit task.
**Figure supplement 1—source data 1.** Pupil slope measures as an index of listening fatigue.

## Predictors of speech intelligibility – shared and private variability

Our overall motivation was to develop objective physiological markers that might explain complaints of poor speech communication in individuals with clinically normal hearing. Here, we examined whether poor speech-in-noise intelligibility was associated with poor auditory nerve integrity (indexed here by ABR wave one amplitude), poor encoding of monaural sTFS cues (as indexed by the FMFR), generally poor fast temporal processing (indexed here by IPDFR) and increased utilization of cognitive resources related to effortful listening (indexed here by pupil change). Importantly, none of these indices were correlated with each other, suggesting that – in principle – each of these markers could account for statistically independent components of the total variance in speech performance (*Figure 5A*, right). In practice, only FMFR and pupil showed a significant independent correlation with speech intelligibility threshold (*Figure 5A*, left).

To determine whether combining these independent metrics could account for an even greater fraction of the total variance, we used a multiple variable linear regression model, and computed the adjusted $R^2$ values, after adding each successive variable. Variables were added in decreasing order of individual $R^2$ values. The adjusted $R^2$ penalizes for model complexity incurred due to the addition of more variables (See Materials and methods). With this model, listening effort indexed by pupil diameter explained 24% of the variance in the digit comprehension task. Adding in monaural sTFS processing measured using the FMFR increased the adjusted $R^2$, explaining 49% of the overall variance. Adding in the ABR wave one provided only a minimal additional increase in predictive power, raising the total explained variance to 52% (*Figure 5B*). Adding additional neural markers such as the IPDFR or extended high frequency thresholds did not provide any further increase in the overall variance explained. Among the neural measures studied here, the best linear model for speech intelligibility included a measure of bottom-up monaural fine structure processing and a measure of top-down listening effort. In order to account for order effects in the model, we also looked at the adjusted $R^2$ for all 2-variable combinations between the FMFR, pupil diameter and the ABR. The combination of FMFR and pupil diameter provided the best model in all order configurations (*Figure 5C*). Finally, even though the behavioral FM detection thresholds and the FMFR were correlated (*Figure 2H*), constructing a model with the psychophysical threshold along with FMFR and pupil diameter increased the variance explained to 78% (*Figure 5B*, gray), suggesting that the behavioral FM detection task reflects additional aspects of auditory processing that is not captured by the combination of peripheral sTFS encoding and non-sensory measures of listening effort.

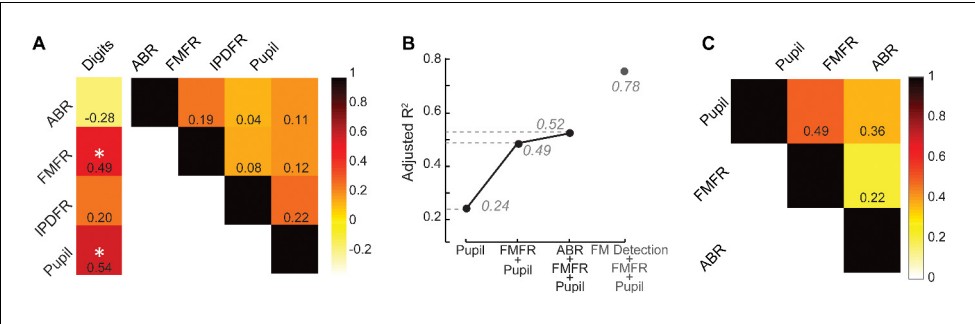

**Figure 5.** A multiple variable linear model of bottom-up and top-down neural and behavioral markers best predicts speech intelligibility thresholds. (**A**) The four neural markers studied here (ABR wave 1, FMFR, IPDFR and pupil-indexed effortful listening) were not correlated with each other. FMFR and pupil were both significantly correlated with the outcome measure, digits comprehension threshold. White asterisk indicates p<0.05 with a univariate linear regression model. (**B**) A multivariate regression model measuring the adjusted $R^2$ (proportion of variance explained by predictors, adjusted for the number of predictors in the model) reveals incremental improvement in the prediction of the digits comprehension threshold when pupil, FMFR and ABR wave one amplitudes are added in succession. Adding additional neural markers to the model did not improve the total explained variance. A separate model which included behavioral FM detection thresholds improved the adjusted $R^2$ value to 0.78 (gray). (**C**) All two-variable combinations were analyzed to study order effects for introducing variables into the model. The combination of pupil diameter and FMFR was still the most optimal model for explaining variance on the speech-in-noise task. Numbers indicate adjusted $R^2$ values for each combination. The online version of this article includes the following source data for figure 5:

**Source data 1.** Digits comprehension thresholds, ABR wave one amplitudes, FMFR slope values, IPDFR slope values and pupil diameter slope values used for the model.

## Discussion

### Neural and perceptual processing of temporal fine structure

The cochlea acts as a limited-resolution filter bank that breaks down the broadband speech waveform into a spatially organized array of narrowband signals. Each cochlear filter contains two types of information that are encoded and reconfigured by neurons within the auditory nerve and central auditory pathway: sTFS and stimulus temporal envelope. STFS cues consist of rapid oscillations near the center of each cochlear filter that are encoded by phase-locked auditory nerve action potentials (*Henry and Heinz, 2013*). Envelope cues, by comparison, reflect slower changes in amplitude over time that can be encoded by the short-term firing rate statistics of auditory nerve fibers (*Joris and Yin, 1992*). The ability to detect slow rates of FM (<~6 Hz) at low carrier frequencies (<~1500 Hz) has long been associated with sTFS processing (*Moore and Sek, 1995*; *Paraouty et al., 2018*; *Sek and Moore, 1995*). Under these stimulus conditions, changes in FM are hypothesized to be conveyed by spike timing information within a cochlear filter (*Moore and Sek, 1995*).

The strong correlation between psychophysical thresholds for detecting pure tone FM and multi-talker speech recognition thresholds is striking (r = 0.85, *Figure 2C*) and has now been documented by several independent groups using a variety of speech-on-speech masking paradigms, but not with non-speech maskers (*Johannesen et al., 2016*; *Lopez-Poveda et al., 2017*; *Strelcyk and Dau, 2009*; *Whitton et al., 2017*). The exact mechanism of FM coding by the auditory pathway is not entirely clear, with some studies suggesting that FM cues are converted to amplitude modulation cues at early stages of auditory processing, and hence that the perception of FM relies more on neural envelope cues (*Ghitza, 2001*; *Whiteford et al., 2017*; *Whiteford and Oxenham, 2015*), while other studies emphasize neural phase-locking to sTFS cues (*Moore et al., 2019*; *Paraouty et al., 2016*; *Wallaert et al., 2018*). The spatial distribution of neural generators for the FMFR also deserves additional study, as some off-channel higher frequency neurons may be combined with low-frequency tonotopically aligned neurons (*Gockel et al., 2015*; *Parthasarathy et al., 2016*).

Here, we characterized processing of FM tones using a combination of classic psychophysical tests and a newly developed ear canal EEG FMFR. Because we looked at changes in phase-locking

to the carrier, and not the actual rate of FM, we were able to explicitly emphasize neural coding of these timing cues in the early auditory system, while minimizing contributions from the recovered envelope, which would be reflected as the 2 Hz FM rate. The FMFR was strongly correlated with behavioral FM detection, suggesting that this response reflects aspects of the behavioral FM detection task (*Figure 2H*), and was also correlated with performance on the digits task (*Figure 2I*), suggesting that the representation of these fine stimulus timing cues contributes to multi-talker speech intelligibility.

Subjects with the lowest FM detection thresholds exhibited a small increase in FMFR amplitudes across a broad range of shallow excursion depths before suddenly increasing at FM excursion depths that exceeded the limits of a single cochlear filter, perhaps indicating the transition from a timing to a place code (*Figure 2F*). By contrast, the IPDFR transfer function in subjects with the lowest IPD thresholds increased steadily for all IPDs above zero (*Figure 3D*). As a result, top psychophysical performers had a shallow transfer function for FM excursion depth but a steep transfer function for IPDFR, producing a positive correlation between FMFR and FM detection threshold (*Figure 2H*) and a negative correlation between the IPDFR and IPD detection threshold (*Figure 3F*). As described above, the FMFR is calculated as the phase coherence to the FM carrier, whereas the IPDFR is calculated as the entrainment to the rate of IPD alternation. As these measures are in no way equivalent, there is no reason to expect the same relationship between each transfer function and the corresponding psychophysical detection threshold.

## Revealing the modes of biological failure underlying hidden hearing disorder

Many forms of cochlear dysfunction could give rise to poor sTFS processing (*Henry et al., 2019*), though when audibility thresholds are normal, the most likely explanation involves a loss of cochlear afferent synapses onto inner hair cells. Auditory nerve fiber loss has been observed in cochlear regions with normal thresholds in many animal species as well as post-mortem analysis of human temporal bone specimens (*Furman et al., 2013*; *Gleich et al., 2016*; *Kujawa and Liberman, 2009*; *Möhrle et al., 2016*; *Valero et al., 2017*; *Viana et al., 2015*; *Wu et al., 2019*). In humans, recent findings suggest that appreciable auditory nerve fiber loss begins in early adulthood, well before degeneration is noted in cochlear sensory cells or spiral ganglion cell bodies (*Wu et al., 2019*). In animal models, a loss of cochlear afferent synapses disrupts temporal coding of amplitude modulation on a variety of time scales without permanently elevating pure tone thresholds, consistent with observations made in our subject cohort (*Bakay et al., 2018*; *Parthasarathy and Kujawa, 2018*; *Shaheen et al., 2015*). In humans, it is impossible to directly assess the status of cochlear afferent synapses in vivo, though indirect proxies for cochlear afferent innervation may be possible (for recent reviews see - *Bharadwaj et al., 2019*; *Bramhall et al., 2019*; *Guest et al., 2019*). Prior work has emphasized the amplitudes of ABR waves and extended high frequency hearing thresholds as possible indirect markers of cochlear synapse loss (*Bharadwaj et al., 2019*; *Garrett and Verhulst, 2019*; *Liberman et al., 2016*). We found considerable individual variability in both of these measures in subjects with clinically normal hearing, although neither measure had a statistically meaningful relationship with multi-talker speech recognition thresholds (*Figure 1J*, *Figure 1—figure supplement 2D*).

Speech perception does not arise directly from the auditory nerve, but rather reflects the patterning of neural activity in the central auditory pathway. Therefore, one might expect a weak correlation between a well-designed proxy for cochlear synaptopathy and a behavioral measure of speech recognition accuracy, but the correlation would never be expected to be too high simply because the periphery is a distal – not proximal – basis for speech perception. Hearing loss profoundly affects gene expression, cellular morphology, neurotransmitter levels and physiological signal processing at every stage of the central pathway - from cochlear nucleus to cortex - and these central sequelae resulting from a peripheral insult would also be expected to affect the neural representation of speech in ways that cannot be accounted for purely by peripheral measures (*Auerbach et al., 2019*; *Balaram et al., 2019*; *Caspary et al., 2008*; *Chambers et al., 2016*; *Möhrle et al., 2016*; *Parthasarathy et al., 2019*; *Sarro et al., 2008*). To this point, the psychophysical FM detection threshold was more highly correlated with speech recognition than the neural measure of low-level FM encoding, suggesting that the behavioral task captured additional aspects of FM detection not present in the FMFR. In a recent placebo-controlled auditory training study, we observed that

thresholds could improve by ~1.5 dB SNR on the same digits task used here without any improvement in FM detection threshold, or any other marker of bottom-up processing, again pointing towards the critical involvement of top-down active listening mechanisms in multi-talker speech perception (*Whitton et al., 2017*). Adding neural markers of higher-order stream segregation to the multivariate model (*Divenyi, 2014*; *Krishnan et al., 2014*; *Lu et al., 2017*; *Shamma et al., 2011*; *Teki et al., 2013*) or direct neural speech decoding (*Ding and Simon, 2012*; *Ding and Simon, 2009*; *Maddox and Lee, 2018*; *Mesgarani et al., 2014*; *Mesgarani et al., 2008*; *Pasley et al., 2012*; *Presacco et al., 2016*) would very likely capture even more of the unexplained variability in multi-talker speech intelligibility, though they offer less insight into the particular mode of sensory failure than FMFR and are also considerably harder to implement in a clinical setting.

Our findings suggest that the individuals who struggle most to follow conversations in noisy, social settings might be identified both by poor bottom-up processing of rapid temporal cues in speech and also by an over-utilization of top-down active listening resources. The interplay between bottom-up and top-down processing is likely more complex than a simple tradeoff where poor sTFS processing is linked to strong pupil-indexed listening effort, or vice versa, as we observed no linear correlation between these variables and no pattern emerged on a subject by subject basis (*Figure 5A*). Understanding how organisms balance bottom-up and top-down processing strategies to encode communication signals is a question of utmost importance (*Enikolopov et al., 2018*; *Mandelblat-Cerf et al., 2014*; *Moore and Woolley, 2019*). In the context of human speech perception, this question would be best tackled by an approach that more explicitly identified the implementation of cognitive active listening mechanisms and was statistically powered to address the question of individual differences (*Jasmin et al., 2019*; *Lim et al., 2019*; *Michalka et al., 2015*).

## Towards clinical biomarkers for hidden hearing disorder

Patients with bilateral normal audiograms represented ~19% of the patient population at the Massachusetts Eye and Ear Infirmary, 45% of whom reported some form of perceived hearing loss as their primary complaint (*Figure 1D*). The combination of an increased lifespan and the increased use of in-ear listening devices will likely exacerbate the societal impact of hidden hearing disorder, leaving hundreds of millions of people straining to follow conversations in noisy, reverberant environments typically encountered in the workplace and social settings (*Goman and Lin, 2016*; *Hind et al., 2011*; *Lin et al., 2011*; *Ruggles et al., 2012*; *Ruggles et al., 2011*). The standard of care at hearing health clinics include measures of pure tone thresholds, otoacoustic emissions, middle ear reflexes and recognition of individual words presented in silence. These tests are useful in diagnosing late-stage hearing loss commonly found with aging, or exposure to ototoxic drugs or intense noise, where there is pathology in the sound transduction machinery of the middle and inner ear. New diagnostic measures and interventions are needed for the silent majority, who struggle to follow conversations in noisy, social environments and avoid seeking clinical care for their hearing difficulties. Here, we present a simple battery of behavioral and physiological tests that can account for nearly 80% of the variability in a test of multi-talker speech intelligibility that does not involve linguistic cues. Low-channel EEG systems and pupillometry cameras are relatively low-cost and could – in theory – be put to use in clinical settings to provide patients with an objective measure for their perceptual difficulties and provide hearing health providers with an objective readout for their therapeutic drugs or devices.

# Materials and methods

## Subjects

All procedures were approved by the institutional review board at the Massachusetts Eye and Ear Infirmary (Protocol #1006581) and Partners Healthcare (Protocol #2019P002423). Twenty seven subjects (13 male, 14 female) were recruited and provided informed consent to be tested as part of the study. Of these, 4 subjects were excluded for either failing to meet the inclusion criteria (1 male, 1 female, see below for inclusion criteria) or not completing more than 60% of the test battery (2 male). One subject (female) did not participate in the electrophysiology tests, but data from the other two sessions were included for relevant analyses. Subjects were compensated per hour for their participation in the study.

## Testing paradigm - Overview

Eligibility of the participants was determined on day of first visit by screening for cognitive skills (Montreal Cognitive Assessment, MOCA > 25 for inclusion), depression (Beck's depression Inventory, BDI <21 for inclusion), tinnitus (Tinnitus reaction questionnaire, TRQ <72 for inclusion), use of assistive listening devices ("Do you routinely use any of the following devices – cochlear implants, hearing aids, bone-anchored hearing aids or FM assistive listening devices' - subjects were excluded if they answered yes to any of the above) and familiarity with English ('Are you a native speakers of English', and 'If not, Are you fluent or functionally fluent in English?' - subjects were excluded if they answered no for both questions). Eligible participants were then tested in a double walled acoustically isolated chamber with an audiometer (Interacoustics AC40, Headphones: TDH39) to confirm normal audiograms with thresholds $\leq$ 20 dB HL for frequencies up to 8 kHz. Participants then performed high frequency audiometry (Headphones: Sennheiser HDA200) and the digits comprehension task paired with pupillometry (described in detail below). Subjects were then sent home with tablet computers (Microsoft Surface Pro 2) and calibrated headphones (Bose AE2). Subjects were asked to complete additional suprathreshold testing (FM detection, IPD detection) and questionnaires - Noise exposure questionnaire (NEQ) (*Johnson et al., 2017*) Speech, spatial and Qualities of hearing scale SSQ (*Gatehouse and Noble, 2014*),Tinnitus handicap questionnaires (*Newman et al., 1996*) in a quiet environment over the course of 8 days. The microphone on the tablet was used to measure ambient noise level throughout home-based testing. If noise levels exceeded 60 dB A, the participant was locked out of the software, provided with a warning about excessive noise levels in the test environment, and prompted to find a quieter location for testing. Subjects returned to the laboratory on Day 10 (±1 day) for electrophysiological testing.

## Speech intelligibility threshold

Subjects were introduced to the target male speaker ($F_0$ = 115 Hz) as he produced a string of four randomly selected digits (digits 1–9, excluding the bisyllabic '7') with 0.68 s between the onset of each digit. Once familiarized, the task required subjects to attend to the target speech steam in the presence of two additional speakers (male, $F_0$ = 90 Hz; female, $F_0$ – 175 Hz) that produced randomly selected digits with matched target-onset times. The two competing speakers could not produce the same digit as the target speaker or each other, but otherwise digits were selected at random. The target speaker was presented at 65 dB SPL. The signal-to-noise ratio of the distractors ranged from 0 to 20 dB SNR. Subjects reported the target 4-digit sequence using a virtual keypad on the tablet screen 1 s following the presentation of the 4[th] digit. Subjects were initially provided with visual feedback on the accuracy of their report in four practice blocks comprised of 5 trials each and 4 SNRs (target only, 20, 9 and 3 dB SNR). Testing consisted of 40 blocks of 8 trials each, with SNRs of 9, 6, 3 and 0 dB presented in a randomized order for each cycle of four blocks. The first three trials of each block served as refreshers to familiarize the subject with the target speaker at 20 dB SNR before progressively decreasing to the test SNR presented in the last five trials of each block. Trials were scored as correct if all four digits entered into the keypad matched the target speaker sequence. The response curve was constructed using percent correct as a function of SNR, and the 70.7% correct point on the response curve was defined as the speech reception threshold. Subjects with thresholds better than 0 dB SNR (n = 4) were marked as 0.

## Frequency Modulation detection threshold

Subjects were introduced to the percept corresponding to frequency modulation (FM) through a virtual slider on the tablet computer that they manipulated to increase and decrease the FM excursion depth of a 500 Hz tone. High excursions were labeled 'squiggly' to allow the subjects to associate the sound with a label that could be used when completing the 2-interval 2-alternative forced choice detection task. After initial familiarization, two tones (carrier frequency = 500 Hz, duration = 1 s, level = 55 dB SL) were binaurally presented with the same starting phase to subjects, with an inter-stimulus interval of 0.5 s. Frequency modulation was applied at a rate of 2 Hz to one of the two tones (order selected at random) and the other tone had no FM. A quasi-sinusoidal amplitude modulation (Amplitude modulation rate randomized between 1–3 Hz, randomized starting phase, 6 dB modulation depth) was applied to both tones to reduce cochlear excitation pattern cues (*Moore and Sek, 1996*). The subject reported whether the first or second tone was 'squiggly' (i.e.,

was the FM tone). A two-down one-up procedure converged on the FM excursion depth that subjects could identify with 70.7% accuracy (*Levitt, 1971*). FM excursion depth was initially set to 75 Hz and was then changed by a factor of 1.5 for the first five reversals, decreasing to a factor of 1.2 for the last seven reversals. The geometric mean of the last six reversals was used to compute the run value. A minimum of 3 runs were collected. The coefficient of variation (standard deviation/mean) for the reversal values was computed during testing. If the coefficient of variation was >0.2, additional runs were collected until this criterion was met or six runs had been collected, whichever came first. The median threshold value obtained across individual runs defined the participant's FM detection threshold.

## Interaural Phase Difference detection threshold

Sensitivity to interaural phase difference was tested using a 2-interval 2-alternative forced choice task. Sound tokens consisted of tones presented simultaneously to both ears at the same carrier frequency (520 Hz), amplitude modulation rate (100% depth at 40.8 Hz), duration (1 s) and level (85 dB SPL, 50 ms raised cosine onset/offset ramps). Each token was separated by a 0.5 s silent interval. Both tokens started in phase. But for one of the two tokens, a phase shift was applied to the tone in each ear in opposing polarity, 0.5 s after tone onset. This produced a perceptual switch, where the sound 'moved' from a diotic to a dichotic percept. The subjects were asked which of two sound tokens 'moved' in the middle. Subjects were familiarized with the task in two practice blocks of ten trials each and provided visual feedback about their accuracy in identifying the tone that 'moved'. A two-down-one-up procedure was used to converge on the phase shift that could be identified with 70.7% correct accuracy. The phase shift was initially set to 81 degrees and changed by a factor of 1.5 for the first four reversals, decreasing to a factor of 1.2 for the last six reversals. The geometric mean of the last six reversals was used to compute the run value. The criteria for determining the number of runs and the threshold matched the FM detection task above.

## Electrophysiology

EEG recordings were performed in an electrically shielded sound attenuating chamber. Subjects reclined in a chair and were instructed to minimize movements. Arousal state was monitored but not regulated. Most subjects reported sleeping through the recordings. The recording session lasted ~3 hr and subjects were given breaks as necessary. Recordings were done on a 16-channel EEG system (Processor: RZ6, preamplifier: RA16PA, differential low impedance amplifier: RA16-LID, TDT Systems) with two foil electrodes positioned in the ear canals (Etymotic) and six cup electrodes (Grass) placed at Fz, Pz, Oz, C7, and both ear lobes, all referenced to ground at the nape. Impedances were kept below 1 kΩ by prepping the skin (NuPrep, Weaver and Co.) and applying a layer of conductive gel between the electrode and the skin (EC2, Natus Medical). Stimuli were delivered using calibrated ER3A (Etymotic) insert earphones. Stimulus delivery (sampling rate: 100 kHz) and signal acquisition (sampling rate: 25 kHz) were coordinated using the TDT system and a presentation and acquisition software (LabView).

Auditory brainstem responses were measured in response to 3 kHz tone pips of 5 ms duration. Stimuli had 2.5 ms raised cosine ramps, and were presented at 11.1 repetitions per second. Presentation level was fixed at 105 dB SPL. Stimulus polarity was alternated across trials and 1000 repetitions per polarity were collected. ABRs from the Fz-tiptrode montage were filtered offline between 300 Hz to 3 kHz. Peaks and following troughs of ABR waves were visually marked by an experienced observer, and wave amplitudes were measured using a peak analysis software (https://github.com/bburan/abr; *Buran, 2019*).

The FMFR was measured in response to sinusoidal FM stimuli with a carrier frequency of 500 Hz, a modulation rate of 2 Hz and at modulation depths of 0 (i.e. a pure tone), 2, 5, 8, and 10 Hz. The stimuli were 1 s in duration with 5 ms raised cosine ramps and presented once every 1.19 s. Stimulus polarity was alternated across trials and 200 samples were acquired for each polarity. The level was fixed at 90 dB SPL. FMFRs from the Fz-tiptrode montage were used for subsequent analyses. Cochlear neural responses to low frequency tones, including the carrier of our FM stimuli, are phase-sensitive in such a way that the summed response to alternating polarities is effectively rectified, and thus periodic at twice the stimulus frequency (*Lichtenhan et al., 2014*). Therefore, we quantified the modulation of the EEG signals with respect to twice the FM carrier frequency, or 1000 Hz. FMFR

amplitudes were calculated using a heterodyne method (*Guinan et al., 2003*). Briefly, a discrete Fourier transform (DFT) was computed for each FMFR average. The negative frequency components were discarded to create an analytic signal. This analytic signal was then down-shifted in frequency so that the components around 1000 Hz became centered at 0 Hz. The frequency-shifted signal was filtered in the frequency domain using an exponential filter (*Shera and Zweig, 1993*), and finally, the inverse DFT was computed. The phase of the inverse DFT is a time-varying signal whose amplitude can be compared directly to the modulation depth of the stimulus. A bootstrapping technique was used to reduce the variability of the calculated FMFR amplitude. An average FMFR was constructed from a subsample of the raw data by drawing 100 samples of each polarity randomly without replacement. This was repeated 1000 times, and the heterodyne analysis was performed on each average. The phase signals output from the heterodyne were averaged and used to compute the final FMFR amplitude. One subject did not yield measurable FMFRs above the noise floor and was excluded from subsequent analyses.

Interaural phase difference following responses were collected to a 520 Hz tone carrier whose amplitude was modulated at 40.8 Hz. The phase of the carrier was modulated to shift in and out of phase at a rate of 6.8 Hz. The amplitude modulation rate and the rate of inter-aural phase shifts were chosen such that the minima of the amplitude modulation coincided with the point of phase shift. The degree of shift per ear was 0° (no shift), 22.5°, 45° and 90°. Presentation level was fixed at 85 dB SPL. Each stimulus condition was presented continuously for 1.47 min, epoched at 294.3 ms to contain 300 epochs with two phase shifts each (one out of phase, and one back into phase) and averaged. IPDFR amplitudes and envelope following response amplitudes were calculated from an FFT performed on the averaged waveforms of the Fz-C7 electrode montage at a resolution equal to 1/epoch length (~3.1 Hz), at 6.8 Hz for the IPD response, and at 40.8 Hz for the AM response. Control recordings consisted of phase shifts of 90° in both ears, but in the same polarity to eliminate the binaural component, which showed no responses at the frequency of the interaural phase shift (6.8 Hz).

## Pupillometry

Task-related changes in pupil diameter were collected with a head mounted pupillometry system at a 30 Hz sampling rate (Argus Science ET-Mobile), while the subjects used a tablet computer to complete the digits comprehension task (Microsoft Surface Pro 2). The dynamic range in pupil diameter was initially characterized in each subject by presenting alternating bright and dark screens via the tablet computer. Ambient light level was then adjusted to obtain a baseline pupil diameter in the middle of the dynamic range. Pupil measurements were made while subjects were instructed to look at a fixation point on the screen during the presentation of the digits, as confirmed by the experimenter with real time gaze tracking. Pupil data were preprocessed to interpolate for blinks and missing periods using a cubic spline fit, outlier values were removed with a Hampel filter, and the signal was smoothed using a 5-point moving average window. Subjects were excluded if they had extended periods of missing data or if the ambient light could not be adjusted to account for excessive dilation. Reliable data was obtained from 16 subjects who were included for subsequent analyses. A single trial included a 2 s pre-trial baseline period, 2 s for presentation of the digit sequences, a 2 s wait period and the presentation of the virtual keypad to indicate their response. The pupil analysis described here comes from the 2 s digit presentation period. Pupil measurements were normalized to the average baseline pupil diameter for each block, collected in the last 500 ms of the baseline period. A single measure of pupil-indexed listening effort was operationally defined as the peak value of the average fractional change in pupil diameter function, calculated independently for each SNR block as ((post-stimulus time point – baseline)/baseline). The amplitude value for each SNR was then expressed as a ratio of the complete dynamic range for each subject to reduce variability due to recording conditions, arousal states and eye anatomy.

## Statistical analysis

The distributions of all of the variables were summarized and examined for outliers. Pairwise linear correlations were computed using Pearson's correlations (r). To assess which sets of predictors best account for variability in our outcome measure (Digits comprehension threshold), all predictors were considered in univariable models and the $R^2$ calculated (SAS v9.4, SAS Institute). Data from the 15

subjects who had reliable pupil and FMFR measures were used in the model building, due to the requirement for a balanced dataset across all metrics. Each predictor was then added to the model from highest to lowest $R^2$, and the adjusted $R^2$ calculated using the formula

$$\text{Adjusted R}^2 = -\frac{(1-R^2)(N-1)}{N-p-1}$$

where $R^2$=sample R-square, p=number of predictors, N = total sample size. The adjusted $R^2$ penalizes for increasing complexity by adding more predictors.

## Acknowledgements

Authors wish to thank William Goedicke and the audiology department of Mass. Eye and Ear for maintaining and providing access to the audiology database. Thanks also to Dr. Jonathon Whitton for help with designing the psychophysical tasks, and Dr. Kelly Jahn for comments on the manuscript.

This study was funded by the National Institutes of Health (NIDCD P50-DC015857) to DBP.

## Additional information

### Competing interests

Kara Bennett: Affiliated with Bennett Statistical Consulting Inc. The author has no financial interests to declare. The other authors declare that no competing interests exist.

### Funding

| Funder | Grant reference number | Author |
| --- | --- | --- |
| National Institutes of Health | P50-DC015857 | Daniel B Polley |

The funders had no role in study design, data collection and interpretation, or the decision to submit the work for publication.

### Author contributions

Aravindakshan Parthasarathy, Investigation, Data curation, Methodology, Formal analysis, Visualization; Kenneth E Hancock, Software, Methodology; Kara Bennett, Victor DeGruttola, Formal analysis; Daniel B Polley, Conceptualization, Supervision, Funding acquisition, Project administration

### Author ORCIDs

Aravindakshan Parthasarathy  https://orcid.org/0000-0002-4573-8004
Daniel B Polley  https://orcid.org/0000-0002-5120-2409

### Ethics

Human subjects: All subjects provided informed consent to be tested as part of the study. All procedures were approved by the institutional review board at the Massachusetts Eye and Ear Infirmary (Protocol #1006581) and Partners Healthcare (Protocol #2019P002423).

### Decision letter and Author response

Decision letter https://doi.org/10.7554/eLife.51419.sa1
Author response https://doi.org/10.7554/eLife.51419.sa2

## Additional files

### Supplementary files

- Transparent reporting form

## Data availability

Data used to plot Figures 1-5, and the supplementary figures are provided as source data files.

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
