## [Decision Letter]

**Acceptance summary:**

This work tackles an important and interesting issue, that is, why subjects diagnosed with normal hearing show difficulty in speech recognition, by systematically assessing the behavioral and physiological measures of both low- (FM tracking) and high-level (pupil response) auditory processing. The study identifies an efficient and robust protocol that could account for large variability (to around 85%) in speech-in-noise performance and thus provides a promising mechanistic interpretation for the "hidden hearing loss" group. Importantly, based on pupil dynamics and ear canal EEG to non-speech FM sounds, they could largely predict the somewhat complex multi-talker speech tracking ability in human subjects. The findings will be of major interest to broad readership, including auditory neuroscientists, clinicians, hearing-aid and cochlear-implant manufactures, and computer scientists who might use AI to diagnose hidden hearing loss.

**Decision letter after peer review:**

Thank you for submitting your article "Neural signatures of disordered multi-talker speech perception in adults with normal hearing" for consideration by *eLife*. Your article has been reviewed by three peer reviewers, one of whom is a member of our Board of Reviewing Editors, and the evaluation has been overseen by Barbara Shinn-Cunningham as the Senior Editor. The following individuals involved in review of your submission have agreed to reveal their identity: Fan-Gang Zeng (Reviewer #2); Christian Lorenzi (Reviewer #3).

The reviewers have discussed the reviews with one another and the Reviewing Editor has drafted this decision to help you prepare a revised submission.

Summary:

This work addresses an important and timely issue, that is, why subjects diagnosed with normal hearing show difficulty in speech recognition("hidden hearing loss"). The study systematically examined peripheral and central performance and identified a combination of behavioral and physiological measures of low- and high-level auditory processing that could largely account for speech-in-noise performance in the "hidden hearing loss" group. All the reviewers agree that it is a timely and carefully performed study and would also have great impacts in a wide range of audience.

Essential revisions:

1) An important implication of the study is to possibly separate the bottom-up and top-down factors that contribute to speech recognition difficulty on a subject-by-subject basis. Could the authors examine the relative ratio of the two factors, e.g., some subjects show more bottom-up contribution whereas other show more top-down limitations?

2) Missing link between the 19% of the 106,787 patients (subsection “Many individuals seek medical care for poor hearing but have no evidence of hearing loss”) and the 23 subjects (subsection “Speech-in-noise intelligibility varies widely in individuals with clinically normal hearing”, first paragraph) who participated in the present study. At first, the reviewers thought these 23 subjects were part of those who complained about hearing difficulty but had normal audiograms, but nowhere can the reviewers find any info to confirm this connection and in fact, the 23 subjects seemed to be independently recruited to meet the audiogram, age, and other criteria. If it was true that those 23 subjects didn't seek any medical intervention because of hearing difficulty, then this fact needs to be explicitly spelled out.

3) The reviewers would strongly encourage the authors to discuss their FMFR data and methods in relationship with a previous study by Gockel et al., 2015, suggesting that FFR for low-frequency pure tones at medium to high levels mainly originates from neurons tuned to higher frequencies.

---

## [Author Response]

Essential revisions:1) An important implication of the study is to possibly separate the bottom-up and top-down factors that contribute to speech recognition difficulty on a subject-by-subject basis. Could the authors examine the relative ratio of the two factors, e.g., some subjects show more bottom-up contribution whereas other show more top-down limitations?

We completely agree; it would be important to understand how bottom-up and top-down resources are leveraged to process low SNR speech, whether their benefits are interchangeable, or whether instead their benefits are complementary and are fully realized when they are both applied together. While this is a question of great importance, it wasn’t the question we set out to answer so one major caveat is that our study design was not powered to fully explore individual strategies. As such, the conclusions we offer below might be better considered as suggestions that could be addressed in a separate study designed with this purpose in mind.

To the reviewers’ question, no, it does not seem that our proxies for bottom-up (FMFR) and top-down (pupil) are interchangeable, such that a subject with strong bottom-up would have weak top-down, or vice versa. This conclusion is based on the following observations:

– No significant correlations are observed between the FMFR and the pupil measures (Figure 5A). They both make contributions towards explained variance in the multivariate model, but the relationship is more complex than a simple tradeoff, where one is strong while the other is weak.

– To address the relationship of each measure with multi-talker speech recognition, we plotted the FMFR and the pupil diameter for each subject and color coded the symbol to represent their speech intelligibility, with darker colors indicated better performance on the digits task. No systematic differences were noted, though the better performers tended to cluster to the bottom left, indicating lesser effort and better FMFRs (Author response image 1, left).

– To label individual subjects according to how they weighted top-down processing relative to bottom-up processing, we sorted the FMFR and pupil values for each subject by ordinal position. We then calculated an ordinal position index defined as (OP_pupil_ – OP_FMFR_)/(OP_pupil_ + OP_FMFR_). An OP index >0 would identifies subjects that rank higher in their utilization of top-down than bottom-up measures, an OP index <0 would indicate a relatively greater emphasis on bottom-up resources, and an index of 0 would indicate a balanced relative utilization of each. We then plotted this OP index as a function of the performance on the digits task but did not find any significant linear correlations. We did find that the subjects with the best multi-talker speech recognition thresholds tended to have stronger bias – in either direction – for utilization of either top-down or bottom-up processing. Subjects with poor speech recognition thresholds all had OP indices closer to 0. (Author response image 1, right).

– Finally, we looked at the multivariate model for the physiological predictors of speech, and plotted the residuals of the model fit. While the residuals cannot make a determination on an individual basis of reliance on bottom up or top down factors, it can identify outliers to the model, i.e. individuals who have an over-reliance on one cue or the other. Only one such outlier was found, and removing this outlier increased the adjusted R2 value of the multivariate model from 0.52 to 0.71. These results suggest that both bottom up and top down factors were equally important in determining the model fit.

**Author response image 1. respfig1:** Relationship between bottom-up and top-down measures and their contributions to multi-talker speech intelligibility.

We have modified the Discussion to address these points:

“Our findings suggest that the individuals who struggle most to follow conversations in noisy, social settings might be identified both by poor bottom-up processing of rapid temporal cues in speech and also by an over-utilization of top-down active listening resources. […] In the context of human speech perception, this question would be best tackled by an approach that more explicitly identified the implementation of cognitive active listening mechanisms and was statistically powered to address the question of individual differences (Jasmin et al., 2019; Lim et al., 2019; Michalka et al., 2015).”

2) Missing link between the 19% of the 106,787 patients (subsection “Many individuals seek medical care for poor hearing but have no evidence of hearing loss”) and the 23 subjects (subsection “Speech-in-noise intelligibility varies widely in individuals with clinically normal hearing”, first paragraph) who participated in the present study. At first, the reviewers thought these 23 subjects were part of those who complained about hearing difficulty but had normal audiograms, but nowhere can the reviewers find any info to confirm this connection and in fact, the 23 subjects seemed to be independently recruited to meet the audiogram, age, and other criteria. If it was true that those 23 subjects didn't seek any medical intervention because of hearing difficulty, then this fact needs to be explicitly spelled out.

We thank the reviewers for identifying something in our original manuscript that may have caused confusion among our readers. We recruited subjects that matched the hearing profiles of subjects in the database, but we did not contact the former patients to be in our study. The clinical audiological assessment of these 23 subjects matched the distribution of our clinical sample and the SSQ questionnaire also identified communication difficulties, particularly in related to multi-talker speech intelligibility showed the greatest variability. For these reasons, the 23 subjects we tested were representative of the sample described from the audiology database. To make this explicit, the revised manuscript has been modified as follows:

“To better understand the nature of their suprathreshold hearing problems, we recruited 23 young or middle-aged listeners (mean age: 28.3+0.9 years) that matched the clinically normal hearing from the database profile (Figure 1—figure supplement 2A). […] Like the patients from the clinical database, the audiograms from these subjects were clinically normal, yet many reported difficulties with speech intelligibility, particularly in listening conditions with multiple overlapping speakers (Figure 1—figure supplement 4A).”

3) The reviewers would strongly encourage the authors to discuss their FMFR data and methods in relationship with a previous study by Gockel et al., 2015, suggesting that FFR for low-frequency pure tones at medium to high levels mainly originates from neurons tuned to higher frequencies.

We consider it unlikely that the FMFR signal arises mainly from neurons tuned to higher frequencies based on data shown in Author response image 2 from pilot experiments on 5 subjects, in which the 500 Hz FMFR was measured with simultaneous noise masking. Whereas the 500 Hz FMFR is totally eliminated with broadband noise (BB Noise) that includes the 500 Hz probe, high-pass (HP) noise ranging from either 640Hz to 25 kHz, 2560 Hz to 25 kHz or 10240 Hz to 25 kHz only moderately attenuated the strength of the following response. This finding suggests that some fraction of the generators may come from high-frequency areas, but that tonotopically aligned low-frequency generators still account for most of the measured signal.

**Author response image 2. respfig2:** Tonotopic contributions to the FMFR revealed using high-pass masking noise.

The Discussion has been modified to include these points:

“The spatial distribution of neural generators for the FMFR also deserves additional study, as some off-channel higher frequency neurons may be combined with low-frequency tonotopically aligned neurons (Gockel et al., 2015; Parthasarathy et al., 2016).”